# Endoscopic Treatment of Rathke’s Cleft Cysts: The Case for Simple Fenestration

**DOI:** 10.3390/brainsci12111482

**Published:** 2022-11-01

**Authors:** Matthias Millesi, Carolyn Lai, Ruth Lau, Vincent Chen Ye, Kaiyun Yang, Matheus Leite, Nilesh Mohan, Ozgur Mete, Shereen Ezzat, Fred Gentili, Gelareh Zadeh, Aristotelis Kalyvas

**Affiliations:** 1Division of Neurosurgery, Toronto Western Hospital/University Health Network, University of Toronto, Toronto, ON M5T 2S8, Canada; 2Department of Neurosurgery, Medical University of Vienna, Vienna 1180, Austria; 3Department of Pathology, University Health Network, University of Toronto, Toronto, ON M5G 2C4, Canada; 4Department of Endocrine Oncology, Princess Margaret Cancer Centre, University of Toronto, Toronto, ON M5G 1X5, Canada

**Keywords:** Rathke’s cleft cyst, endoscopy, simple fenestration

## Abstract

Background: Rathke’s cleft cysts (RCC) arise from the pars intermedia because of incomplete regression of the embryologic Rathke pouch. A subset of RCC becomes symptomatic causing headaches, visual and endocrinological disturbances such that surgical intervention is indicated. Several points in surgical management remain controversial including operative strategy (simple fenestration (SF) vs complete cyst wall resection (CWR)) as well as reconstructive techniques. Methods: A retrospective analysis was conducted of pathologically confirmed RCC operated on by endoscopic endonasal approach from 2006 to 2019. Pre-operative symptoms, imaging characteristics, operative strategy, symptom response, complications and recurrences were recorded. Results: Thirty-nine patients were identified. Thirty-three underwent SF and six underwent CWR. Worsening pituitary function was significantly increased with CWR (50%) compared to SF (3%) (*p* = 0.008). All patients underwent “closed” reconstruction with a post-operative CSF leak rate of 5% (3% SF vs 16% CWR, *p* = 0.287). Six (15%) recurrences necessitating surgery were reported. Recurrence rates stratified by surgical technique (18% SF vs 0% CWR, *p* = 0.564) were not found to be significantly different. Conclusions: The current series illustrates variability in the surgical management of RCCs. SF with closed reconstruction is a reasonable operative strategy for most symptomatic RCCs cases while CWR can be reserved for selected cases.

## 1. Introduction

Rathke’s cleft cysts (RCC) grow formed from remnants of the embryologic Rathke’s pouch, and are encountered with a prevalence of approximately 4% [1]. Typically, they represent benign nonneoplastic cysts in the sellar or suprasellar area [2]. Most of these lesions are asymptomatic and remain stable for a long period of time [2,3]. However, a subset of these RCC can become symptomatic or show an increase in size [3]. These associated symptoms include headaches, visual disturbances consisting of visual field deficits or deterioration in visual acuity due to displacement and compression of the surrounding critical structures such as the optic chiasm [1]. Furthermore, endocrinological disturbances with hypopituitarism are also possible, resulting from the compression of the pituitary gland [3]. Some RCCs can present with symptoms similar to pituitary apoplexy due to rupture and leakage of fluid content including sudden onset of headache, cranial nerve palsies, visual decline and again endocrinological disturbances [4]. Accompanying histological changes with xanthomatous reactions may underly these features [5].

With the advent of more advanced neuroimaging, the frequency of incidentally discovered RCCs has increased [6]. Most of these lesions that are encountered incidentally and do not cause any symptoms can be managed conservatively with serial neuroimaging [2]. Surgical treatment is usually indicated only in such symptomatic or growing RCCs [2]. Typically, growing or symptomatic RCCs are treated via a trans-sphenoidal approach [2]. Similar to other sellar lesions and also other intracranial cystic lesions in adults and pediatric patients, the endoscopic techniques have become the preferable option by most institutions in the last decades [7,8].

In this sense, surgical decompression for the relief of associated symptoms can be achieved by simple fenestration (SF) of the RCC. However, complete resection of the RCC and its wall has been described in the existing literature [2]. The aim of this study is to evaluate the results of surgical decompression of a series of RCCs via a fully endonasal endoscopic approach and compare available surgical techniques and their correlation with various outcomes.

## 2. Methods

### 2.1. Patient Recruitment

All patients that underwent fully endoscopic endonasal transsphenoidal treatment of a histopathologically confirmed RCC at the Toronto Western Hospital, University Health Network since 2006 were included. Approval of this study from the local Ethics Committee was obtained and an informed consent was waived due to the retrospective nature of this study.

### 2.2. Data Collection

Of all included patients, the clinical notes, letters and operative notes were reviewed from the electronic data management system. For all patients, the major presenting symptoms and the indication for surgery were noted. In addition, the operative report was specifically analyzed for steps for RCC fenestration, removal of the cyst wall if performed, occurrence of any intraoperative complications and the steps for skull base reconstruction. Furthermore, all potential complications were documented during the postoperative period.

### 2.3. Imaging

The neuroimaging studies consisted of magnetic resonance imaging (MRI) including fluid-attenuated T2-weighted and T1-weighted sequences as well as thin-sliced multiplanar reconstructible (MPR) T1-weighted sequences for intraoperative surgical navigation purposes. 

From these, characteristics such as the largest diameter and the volume of the RCC (calculated by the approximation formula A × B × C/2) or radiological signs of hemorrhage were documented. In addition, the overall appearance was noted as either cystic or mixed solid/cystic. The first postoperative MRI was typically performed before the 3 months postoperative visit.

### 2.4. Visual and Endocrinological Examination

Visual acuity (VA) and visual field (VF) examinations were performed in each patient pre- and postoperatively. Furthermore, a full endocrinological assessment was performed preoperatively, in the first postoperative day and 3 months postoperatively.

### 2.5. Follow-Up and Recurrence

After the first postoperative MRI (approximately 3 months after surgery), follow-up visits including MR-imaging together with VF, VA as well as an endocrine assessment were performed annually. Recurrence of an RCC was defined as cyst re-accumulation that led to retreatment due to serial growth or reoccurrence of symptoms. Timing of recurrence was calculated from time of surgery to the first MRI showing re-accumulation of the RCC.

## 3. Result

### 3.1. Descriptive Data

The overall study cohort consisted of 39 patients in whom an endoscopic endonasal transsphenoidal treatment of an RCC was performed since 2006. In all cases, the diagnosis of an RCC was histopathologically confirmed and therefore, only these cases represented the final study cohort. Mean age at diagnosis was 48 years (range 17–76 years) and the female to male ratio was 2.1:1. Of all included patients, 33 cases presented with newly diagnosed RCCs (85%) whereas 6 patients previously underwent treatment of an RCC and presented with recurrence (15%). A complete list of all patients’ characteristics is shown in Table 1.

### 3.2. Clinical Presentation and Indication for Surgical Treatment

In five of the six patients that underwent treatment for their re-accumulating RCC, a growing lesion was observed. Additionally, two of these reported visual disturbances together with their RCC recurrence. Furthermore, one patient presented with symptoms of a ruptured RCC. Of all newly diagnosed patients with an RCC, the most commonly reported symptoms were headaches in 16 patients (48%) followed by visual disturbances in 13 cases (39%). Acute onset with symptoms of cyst rupture was the clinical presentation in four patients with newly discovered RCCs (12%). For a complete list of all symptoms see Table 1.

### 3.3. Preoperative Work-Up

In their routine preoperative assessment with neuro-ophthalmological examinations, 20 patients (51%) showed deficits in VF testing. More specifically, a bitemporal hemianopia was present in 10 patients (25%) whereas a quadrantanopia was observed in 5 patients (13%) and peripheral scotomas in an additional 5 cases (13%). In one patient, the detailed description of the VF deficit was missing (3%). Furthermore, 19 patients showed a reduced VA (49%) upon examination.

A normal pituitary function was observed in 22 patients (56%). In contrast, 7 patients presented with partial hypopituitarism (18%) and 10 patients with panhypopituitarism (26%). Moreover, three patients showed diabetes insipidus (DI) preoperatively. A complete overview of the preoperative work-up is given in Table 1.

### 3.4. Radiological Examinations

Preoperative MRI was available for analysis in 37 of the 39 patients. The average maximum diameter of all RCCs was 21 mm (range 10–39 mm) and the mean calculated volume was 3.17 mL(range 0.45–13.00 mL). RCCs appeared as a simple cystic lesion in 33 of cases (89%). In contrast, four of the included RCCs were classified as mixed solid/cystic (11%). Signs of intralesional hemorrhage were observed in five patients (14%). Furthermore, five patients (14%) showed signs of swollen sphenoidal mucosa with features of sinusitis in preoperative imaging.

### 3.5. Endonasal Endoscopic Procedures and Perioperative Complications

All 39 patients underwent endonasal endoscopic surgery for fenestration of the RCC. SF, partial (subtotal) resection of the anterior aspect of the cyst wall and fenestration of the RCC was performed in 33 of these cases (85%), whereas a more radical removal of the cyst wall was performed in 6 patients (15%). Of patients in whom a radical cyst wall removal (CWR) was performed, all were initially suspected of harboring a craniopharyngioma and thus a more aggressive treatment approach was chosen. Examples of pre- and postoperative imaging in patients that underwent SF or CWR are given in Figure 1. During surgery, a notable cerebrospinal fluid (CSF) leak occurred in 11 patients (28%).

Following fenestration of the RCC or resection of the cyst wall, closed skull-base reconstruction was performed using mucosal flap harvested from the middle turbinate plus gel-foam cubes and fibrin glue in five patients (13%) who represented the initial cases. In the remaining 34 patients (87%), a multilayered skull base reconstruction using a dural substitute plus/minus nasoseptal flaps harvested at the nasal phase of the procedure was performed.

In the perioperative period, no mortalities occurred, and postoperative complications were noted in four patients (10%). These included a CSF leak in two cases (5%) and one of these patients required a ventriculo-peritoneal shunt insertion for recurrent CSF leaking. Additionally, two patients (5%) developed delayed epistaxis which resolved with packing in one patient and one patient required surgical exploration and cauterization of the sphenopalatine artery.

Analyzed by the applied surgical technique, a higher rate of postoperative complications was observed if CWR was performed (two of six cases, 33%) compared to only SF (2 of 33 cases, 6%). However, this difference did not reach statistical significance (*p* = 0.104). Similarly, if only the occurrence of postoperative CSF leaks was analyzed (*n* = 2 cases), a higher rate could be observed between CWR (one case, 17%) and SF (one case, 3%) but this difference was not statistically significant (*p* = 0.287).

### 3.6. Postoperative Outcome, Changes in Visual Function and Endocrinological Outcome

Of all 16 patients who reported headaches preoperatively, information on outcome was available in 15 cases (94%). Of these, headache improved or resolved postoperatively in 13 patients (81%).

An improvement in VF deficits was noted in 15 patients (38%), whereas stable results from VF testing were seen in 21 cases (54%). In contrast, one patient (3%) showed deterioration with a new bitemporal hemianopia postoperatively that also did not resolve on further follow-up. Data on VF outcome were missing in two patients (5%). Regarding assessment of VA, improvement was noted in 11 patients (28%), whereas 24 patients remained stable in their VA examinations (62%). No patient showed a worsening of VA following surgery. Postoperative examinations of VA were missing in four patients (10%). For an overview of postoperative outcomes, see Table 2.

Compared to the preoperative assessment, a worsening of pituitary function was observed in four patients (10%). Of these, three patients underwent more radical CWR instead of SF (one patient). This difference was statistically significant (*p* = 0.008) (see Table 3).

Overall, seven cases showed permanent DI (18%) in the follow-up period. Of these, three patients had required substitution for DI already preoperatively. As such, newly developed DI occurred in four patients (10%). This occurred in two cases (6%) of the SF technique and two cases (33%) of the CWR technique. This difference in the frequency did not reach statistical significance (*p* = 0.104) (see Table 3).

### 3.7. Follow-Up and Recurrence

For all patients, a mean follow-up time of 58 months was noted (range 3–156 months). Re-accumulation of the RCC requiring reoperation was seen in six patients (15%). A comparison to the performed procedure did not reveal a statistically significant difference (18% for SF vs. 0% for CWR; *p* = 0.564) although a trend is apparent with no recurrences noted in patients that underwent resection of the cyst wall (see Table 3).

## 4. Discussion

### 4.1. Epidemiology and Presentation

RCCs are sellar or suprasellar nonneoplastic cystic lesions, generally found in adults, arising from remnants of Rathke’s pouch. Symptomatic cases are rare and the vast majority are incidental [9]. When they grow, they can compress the surrounding structures and become symptomatic. Headaches, present in 33–81% of patients, represented the most common symptom in this cohort [10,11]. However, despite the fact that some characteristics have been described for headache associated with RCC, there is no pathognomonic feature [12]. Hence, we cannot be sure that this symptom was definitely related to the RCC; however, if postoperative improvement was noted, such a relation was assumed. This symptom was followed by visual disturbances (affecting 12–58% of patients) [6,13] and pituitary hormone abnormalities (one or more axes affected in 19–81% of patients) [3]. Some RCCs can present with symptoms similar to pituitary apoplexy due to rupture of the cyst and leakage of fluid, and may or may not be associated with intracystic hemorrhage and pathological findings of xanthomatous hypophysitis, as shown in one of our institutional series [5]. In these cases, symptoms include sudden onset or increase in severity of headache, with visual disturbance, nausea and vomiting, meningismus, oculomotor palsies, diplopia, impairment of pituitary function [14,15] and potential alterations in mental status [14].

Macroscopically, the cyst is encapsulated, and the contents are generally thick, mucoid or gelatinous material [6,9]. For non-symptomatic cysts, monitoring with serial brain imaging is the usual management. However, when cysts are large enough to manifest clinical symptoms, surgery can be considered [16]

The authors present here a single-center cohort of RCCs, surgically managed by an endoscopic endonasal transsphenoidal approach with long term follow-up. The most common symptom at presentation was headaches (48%), followed by visual disturbances (39%) and pituitary dysfunction (44%).

### 4.2. Surgical Strategy: Simple Fenestration vs GTR

Typically, an endonasal transsphenoidal approach is the applied surgical corridor for most growing or symptomatic RCCs. In the last few decades, the endoscopic techniques have become the preferred option in many centers. The endonasal route allows direct opening and decompression, nevertheless, certain anatomical aspects need to be addressed if this access is chosen; in fact, a concha bullosa can be encountered narrowing the actual corridor which then can be widened by e.g. one-sided middle turbinectomy [17]. Similarly, septal spurs can be difficult to surpass with the endoscope or the instruments and sometimes need correction. If a nasoseptal flap is harvested for reconstruction, the septal spur is exposed as part of the technique and then can easily be removed [17]. The optimal surgical strategy remains controversial [18]. Operative strategies for RCCs range from simple cyst fenestration, marsupialization, silastic spacing, steroid-eluding stenting to partial or complete cyst wall and gross total resection [19,20,21]. 

Most studies advocate for fenestration and aspiration of cyst contents +/− partial excision of the cyst wall [19,22,23]. Although a theoretical risk of recurrence exists, Fan et al found that gross total resection did not reduce the recurrence rate but increased the risk of postoperative complications [24]. This is echoed by Barkhoudarian et al who found that simple cyst drainage had good rates of improvement in pituitary gland function, visual function and headache resolution with low complication rates and symptomatic recurrence risk [25]. 

However, there are recent reports from institutions recommending gross total resection of the cyst wall, documenting satisfactory surgical results with low complications and recurrence rates, stating that radical excision does not necessarily result in endocrinologic impairment and may have a positive impact on reducing recurrences [2,18]. 

Reflecting on the outcomes of our large cohort, we resonate with most groups that SF is an effective and frequently sufficient surgical strategy to treat RCCs. It balances the small risk of symptomatic recurrence over the risk of complication, and in particular, pituitary dysfunction which we experienced in our cohort. At our institution, 33 patients were treated with SF. However, in contrast, six patients were treated with CWR. The decision to perform a more extensive procedure in these cases was based on the potential differential diagnosis of craniopharyngioma that had been raised as this remains challenging in sellar and suprasellar lesions [26,27]. However, our observation was that those cases in whom uncertainty was raised underwent imaging in smaller community-based hospitals. Hence, our strategy to reduce this now includes in-house preoperative MR-imaging and evaluation by subspecialized radiologists. Furthermore, preoperative computed tomography to detect potential calcifications can also help to distinguish these lesions.

Stratified by type of surgical intervention, worsening pituitary function was found to be statistically significantly increased with CWR (50% of cases) compared to SF (3% of cases, *p* = 0.008). The overall rate of postoperative complications was higher in cases with attempted CWR (33% of cases) compared to SF (3%), however, not statistically significant (*p* = 0.104). Similarly, new postoperative DI and CSF leak were also found to be increased with CWR, however, not reaching statistical significance (*p* = 104 and *p* = 0.287, respectively). Re-accumulation of RCC requiring reoperation was seen at a rate of 15%. Although the difference between the two surgical techniques did not reach statistical significance (*p*= 0.564), the recurrence rate was clearly higher in the SF (18%) compared to the CWR group (0%). 

### 4.3. Clinical and Endocrinological Outcome

After surgical treatment, improvement/resolution of headaches has been reported between 44% to 96% [2,28]. Similarly, in our cohort, 81% of patients reported resolution or improvement in headaches after surgery. 

Improvement in the VF was seen in 38% of cases and only one case (3%) showed a deterioration in the VF after surgery. This particular patient was analyzed in detail to identify a potential cause for this clinical deterioration. We observed, in the pre-operative MRI, that the lesion had caused a significant displacement of the optic nerves in an anterior–inferior direction which is not typical in these lesions (Figure 2). As a result, we encountered the optic apparatus immediately after exposure and opening of the dura. This might represent a risk factor for optic nerve injury and demonstrates that the meticulous study of the pre-operative imaging cannot be overemphasized. 

Improvement in VA was noted in 28% of patients. Overall, 49% of patients experienced improvement in their visual disturbances after surgery, which is similar to the 48% of improvement reported in other series [28]

Regarding the endocrinological outcome and integrity of the pituitary gland, 10% of patients experienced a postoperative worsening in their pituitary function. Of these four patients, three of them underwent CWR, while one had had SF, as stated before. Despite the small numbers, this difference was statistically significant (*p* = 0.008). Hypogonadism was present in only one case (3%) of the patients that underwent SF, in agreement with other series reporting a lower rate of hypogonadism with SF (2.5%, reported by Aho et al.) [29]. 

Furthermore, we observed four cases with newly developed permanent DI in the postoperative period (10%), an outcome reported around 5% but that seems to depend on the surgical strategy applied [19,20]. Again, as outlined above, a higher frequency of newly developed permanent DI was observed in the group that underwent CWR (33% vs. 6%). However, this difference did not reach statistical significance most likely due to the small numbers (*p* = 0.104). Similarly, in the literature, new postoperative permanent DI was associated with the attempted gross total resection of the cyst [19,20].

### 4.4. Reconstruction 

Various skull-base reconstruction strategies can be employed in the endoscopic treatment of RCC. The aims include balancing prevention of recurrence and CSF leak. The strategies include leaving the fenestrated cyst “open” or closing the cavity with various reconstruction materials and/or the use of nasoseptal flaps, particularly when intraoperative CSF leak is encountered [30,31]. In this series, all cases underwent closed reconstruction with dural substitute, gelfoam, tissue sealant +/− the use of nasoseptal flap. By this approach, a 5% post-operative CSF leak rate (2/39 cases) was encountered (despite a 25% intraoperative CSF leak rate). One was in the cyst wall resection group (17%) and was associated with a complicated course requiring VP shunt insertion for recurrent CSF leaking, the other was in the cyst fenestration technique group (3%).

Post-operative CSF leak rates range from 0–13% and studies often do not report specific methods of reconstruction [19,22,30]. When reconstruction details are provided, CSF leak rates are not stratified based on specific methods of reconstruction [31]. The use of nasoseptal flaps particularly for larger RCCs or those where intraoperative CSF leak occurs has become more prevalent especially in more recent years [21,32]. Kuan et al. reported a series using a circumferential nasoseptal flap lining the RCC cavity to prevent re-epithelization of marsupialized RCCs [21]. In their series, no patients had radiographic or clinical recurrence or CSF leak [21]. The authors feel that the use of reconstruction with adjuncts including nasoseptal flaps is an effective strategy to minimize CSF leak rates with a negligible risk of symptomatic recurrence and minimal added morbidity [33]

In the literature, there is a general postulation that the reconstruction of the sellar cavity after fenestration of the cyst wall may precipitate the recurrence of the cyst. However, the meta-analysis conducted by Mendelson et al., [19] showed that while two papers [34,35] concluded that reconstruction of the sellar floor and packing of the sellar cavity was a risk factor for recurrence, five other papers [34,36,37,38,39] did not find increased recurrence rates after reconstruction. Consistent with the reported literature, in our series of cases, we did not find that closed reconstruction of the sellar floor was a conclusive factor for developing recurrence.

### 4.5. Recurrence

The recurrence rate requiring reoperation in our cohort was 15% (six cases), in keeping with the literature ranging from 0% in small series to as high as 30% [18,40]. A systematic review of 1151 cases found an overall recurrence rate of 12.5%, with no conclusive evidence that more aggressive resection resulted in lower rates of recurrence [19]. In the recent cohort described by Barkhoudarian in 2019, 22% of patients had some cyst reaccumulation on follow-up MRI while 8% were symptomatic requiring reoperation which was uneventful [33]. Lin et al., in a large (*n* = 109) cohort of RCC cases treated with SF, reported appreciable recurrence rate warranting serial follow up; although usually asymptomatic, 19% required reoperation [28]. There is further evidence to suggest that factors other than surgical technique influence recurrence and should be taken into account. A systematic review found rates of post-surgical recurrence to range from 16–18% and to be associated with factors such as suprasellar location, superinfection of the cyst, use of a fat graft into the cyst cavity, inflammation and reactive squamous metaplasia in the cyst wall [41]. The latter two factors have also been associated with a higher proportion of reoperation in a large case series [42].

For recurrent cases and especially following multiple recurrences, radiation treatment and stereotactic radiosurgery have been described in the literature as another treatment option [43,44]. However, given the typically benign histology of these lesions, we advocate the performance of a second surgery for a recurrent lesion before considering other treatment options. None of the included cases that underwent surgery for their recurrence have required further treatment so far.

## 5. Limitations

This study has several important limitations. First, its retrospective design comes with inherent biases and the evaluation of electronic patient charts can lead to potential dropouts and missing data. However, by our approach of cross-checking the histopathological diagnoses, we identified all consecutive cases in which a definitive diagnosis was available. Similarly, by the centralized referral pattern at our institution, dropout rates during follow-up are at a very low number. Moreover, given the rarity of these lesions, the small number of included patients limits the overall validity of the statistical calculations. Nevertheless, despite the small numbers, important and statistically significant differences could be observed in our cohort. Furthermore, no pathognomic feature for headache associated with an RCC exists, so the relation of this symptom to the pathology was assumed in case of postoperative improvement only.

## 6. Conclusions

The endoscopic endonasal approach represents a straightforward surgical technique for achieving cyst opening and decompression in symptomatic or growing RCCs. The best surgical strategy, cyst fenestration versus radical resection of the cyst wall is still controversial. Based on our data, we advocate for SF as we observed a significantly lower rate of postoperative hypopituitarism while we did not record a statistically significant higher rate of recurrence compared to CWR.

## Figures and Tables

**Figure 1 brainsci-12-01482-f001:**
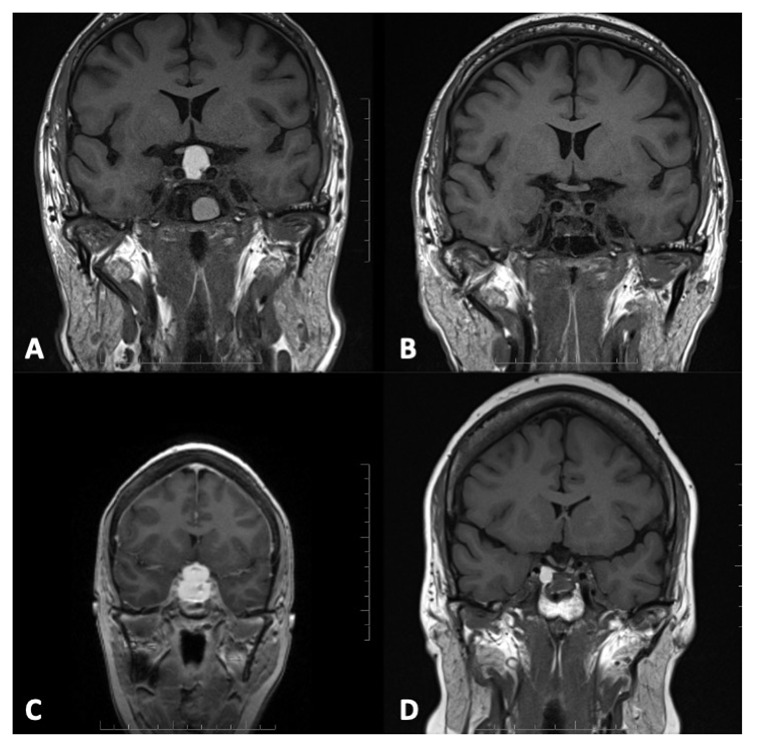
Preoperative (**A**) and postoperative (**B**) T1-weighted MRI of an illustrative case of a patient with suprasellar extension and chiasmal compression of a RCC that underwent CWR is shown in the upper row of the figure. In the lower part, preoperative (**C**) and postoperative (**D**) T1-weighted MRI also depict an illustrative patient with a Rathke’s cleft cyst in whom SF was performed.

**Figure 2 brainsci-12-01482-f002:**
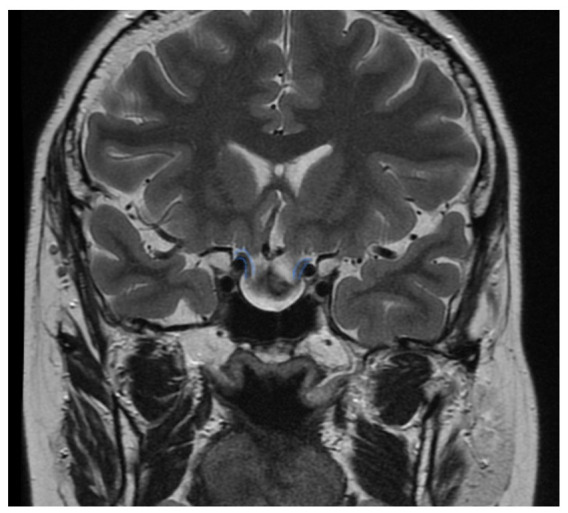
This T2-weighted MR-image depicts the unusual position (blue lines) of the anteriorly-inferiorly displaced optic nerves in the patient that experienced a new bitemporal hemianopia postoperatively.

**Table 1 brainsci-12-01482-t001:** Patient characteristics.

	*n*	%
mean age at diagnosis (range)	48 years (17–76 years)	
female to male ratio	2.1/1	
newly diagnosed Rathke’s cleft cyst (RCC)	33	85
recurrent RCC	6	15
symptoms in newly diagnosed patients		
headache	16	48
visual disturbances	13	39
symptoms of pituitary dysfunction	6	18
rupture of the RCC	4	12
diabetes insipidus	1	3
incidental finding	1	3
indication for surgery in patients with recurrent RCCs		
growth of RCC	3	50
visual disturbances	2	33
pituitary apoplexy	1	17
visual field examination		
normal	18	46
peripheral scotomas	5	13
quadrantanopia	5	13
bitemporal hemianopia	10	25
missing	1	3
visual acuity assessment		
normal	20	51
reduced	19	49
preoperative status of the pituitary gland		
normal	22	56
partial hypopituitarism	7	18
panhypopituitarism	10	26
neuroimaging		
mean diameter of RCC (range)	21 mm (10–39 mm)	
mean volume (range)	3.17 mL (0.45–13.00 mL)	
cystic appearance		
cystic	33	89
mixed cystic/solid	4	11
radiologic signs of hemorrhage	5	14
radiologic signs of sinusitis	5	14

RCC: Rathke’s cleft cyst.

**Table 2 brainsci-12-01482-t002:** Postoperative outcome.

	Postoperative Outcome			
	Improved (%)	Stable (%)	Worse (%)	Missing (%)
preoperative visual fields				
normal	0	17	1	0
peripheral scotomas	3	1	0	1
quadrantanopia	4	1	0	0
bitemporal hemianopia	7	2	0	1
missing				1
preoperative visual acuity				
normal	0	20	0	0
reduced	11	4	0	4
preoperative pituitary function				
normal	0	18	4	0
partial hypopituitarism	0	7	0	0
panhypopituitarism	0	10	0	0
permanent diabetes insipidus (DI)				
no deficit preoperatively	0	36	4	0
preoperative DI	0	3	0	0

**Table 3 brainsci-12-01482-t003:** Comparison of outcome to type of surgical intervention.

	Type of Surgical Intervention		
	Simple Fenestration	Cyst Wall Resection	*p* Value
worsening of pituitary function	3%	50%	***p* = 0.008**
new postoperative diabetes insipidus	6%	33%	*p* = 0.104
postoperative complications	6%	33%	*p* = 0.104
Postoperative cerebrospinal fluid leak	3%	17%	*p* = 0.287
re-accumulation of Rathke’s cleft cyst	18%	0%	*p* = 0.564

## Data Availability

The data presented in this study are available upon request from the corresponding author. The data are not publicly available due to patient confidentiality.

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
