# Peer review of "Endoscopic Treatment of Rathke’s Cleft Cysts: The Case for Simple Fenestration"

_brainsci, 2022, doi:10.3390/brainsci12111482_

Round 1

Reviewer 1 Report

minor corrections

- the authors used the template of the journal?

- A wide variety of intracranial cysts is known to occur in infants. If symptomatic, they require treatment; the ideal surgical treatment and indications of surgery are yet a matter of discussion. Traditional treatment is either by cystoperitoneal shunting, or microsurgical fenestration. Endoscopic treatment is an alternative procedure that avoids the invasiveness of open craniotomy and the complications caused by shunting. , please discuss and cite doi:10.1007/s00381-021-05264-y

- endoscopic nasosinusal surgery can find very frequent nasosinusal structural abnormalities that need surgical correction so that the volume occupied by concha bullosa or septal spurs that do not allow visualization of nasopharyngeal structures can be reduced, please discuss and cite doi:10.1007/s00405-021-06724-6

Author Response

We are very thankful for this reviewer’s comments and tried to respond to the suggestion as detailed as possible:

- the authors used the template of the journal?

Thank you for pointing out this aspect. We formatted the revised version of the manuscript according to the template.

- A wide variety of intracranial cysts is known to occur in infants. If symptomatic, they require treatment; the ideal surgical treatment and indications of surgery are yet a matter of discussion. Traditional treatment is either by cystoperitoneal shunting, or microsurgical fenestration. Endoscopic treatment is an alternative procedure that avoids the invasiveness of open craniotomy and the complications caused by shunting. , please discuss and cite doi:10.1007/s00381-021-05264-y

We agree with this reviewer’s comment that a wide variety of intracranial cysts in adult and pediatric patients are treated with endoscopic techniques nowadays. We therefore added this information and the suggested citation in the introduction section of the revised version of the manuscript.

- endoscopic nasosinusal surgery can find very frequent nasosinusal structural abnormalities that need surgical correction so that the volume occupied by concha bullosa or septal spurs that do not allow visualization of nasopharyngeal structures can be reduced, please discuss and cite doi:10.1007/s00405-021-06724-6

We agree with this reviewer’s suggestion about nasal abnormalities and potential obstacles limiting the view during the approach. Therefore, we added some details on the surgical approach and the suggested reference in the revised version of the manuscript.

Reviewer 2 Report

I wish you good work

Author Response

We are very thankful for this reviewer encouraging words

Reviewer 3 Report

Your article is eloquent and methodic, the topic chosen is interesting and fully debated. Please, find below some considerations:

-       Pre and post-operative imaging should be added, as example, for both procedures

-       Please, use full names before abbreviations (i.e. Table 2 “DI”)

-       Discussion is very confused and mixed with all literature studies reported. I suggest a careful revision, aiming to enhance your point of view.

-       What is the novelty? This is an already debated topic. Please, enhance your point of view and what you want to add with your case series.

-       Conclusions are too confused and repetitive, they need to be revised, especially: “Based on our data, we advocate for SF as we did not see a statistically significant higher rate of recurrence. However, we observed a significantly higher rate of worsening of pituitary function when a radical resection of the cyst wall was attempted.” Clarify your thoughts.

Author Response

We are very thankful for this reviewer’s comments and tried to respond to the suggestion as detailed as possible:

Your article is eloquent and methodic, the topic chosen is interesting and fully debated. Please, find below some considerations:

-       Pre and post-operative imaging should be added, as example, for both procedures

We agree with this reviewer’s comment that this is an important aspect for the understanding of the procedures and therefore added pre- and postoperative MRI-scans for both procedures in the revised version of the manuscript.

-       Please, use full names before abbreviations (i.e. Table 2 “DI”)

Thank you for pointing out this aspect. We thoroughly went through the revised version of the manuscript and added the full names before the abbreviation.

-       Discussion is very confused and mixed with all literature studies reported. I suggest a careful revision, aiming to enhance your point of view.

We are thankful for pointing out the structure of the discussion. We carefully reviewed this section and shortened it in the revised version of the manuscript.

-       What is the novelty? This is an already debated topic. Please, enhance your point of view and what you want to add with your case series.

We agree with this reviewer’s comment that this topic has been debated in the existing literature. Yet, uncertainty about the best treatment approach and technique. Therefore, we wanted to add to the existing literature by providing a simple yet effective method for decompression of growing or symptomatic Rathke’s cleft cysts

-       Conclusions are too confused and repetitive, they need to be revised, especially: “Based on our data, we advocate for SF as we did not see a statistically significant higher rate of recurrence. However, we observed a significantly higher rate of worsening of pituitary function when a radical resection of the cyst wall was attempted.” Clarify your thoughts.

We are thankful for pointing out this aspect in the original manuscript and reviewed the conclusions in the revised version carefully in order to clarify the message that we advocate for a simple yet effective surgical decompression by simple fenestration.

Reviewer 4 Report

The authors have reported the surgical outcomes of endonasal endoscopic surgery for Rathke’s cleft cysts, focusing on simple fenestration (SF) versus complete cyst wall resection (CWR).

The discussion is also detailed, and I consider the paper to be highly complete.

We would appreciate if you could correct the following points.

  1. The authors need to explain how they determined that the headache symptoms were tumor-related.
  2. Please discuss how to reduce the number of Rathke’s cleft cysts operated on as craniopharyngiomas in the future.
  3. CWR is omitted in some places and spelled out in full in others. Please check.
  4. Are there any examples of radiotherapy at recurrence? Also, if you have researched radiotherapy, you may want to include it in the discussion,

Author Response

We are very thankful for this reviewer’s comments and tried to respond to the suggestion as detailed as possible:

The authors have reported the surgical outcomes of endonasal endoscopic surgery for Rathke’s cleft cysts, focusing on simple fenestration (SF) versus complete cyst wall resection (CWR).

The discussion is also detailed, and I consider the paper to be highly complete.

We would appreciate if you could correct the following points.

  • The authors need to explain how they determined that the headache symptoms were tumor-related.

We are grateful for this comment as we absolutely agree that it is difficult to determine whether these symptoms are associated with the Rathke’s cleft cyst. We therefore added this aspect in the revised version of the manuscript and pointed this out also in the limitations section.

  • Please discuss how to reduce the number of Rathke’s cleft cysts operated on as craniopharyngiomas in the future.

We are thankful for this reviewer’s comment as this is of utmost importance for surgical planning. Therefore, we changed the revised version of the manuscript to discuss this aspect as we advocate to improve imaging including reviewing with an subspecialized radiologist.

  • CWR is omitted in some places and spelled out in full in others. Please check.

We are thankful for pointing out this aspect and carefully reviewed it in the revised version of the manuscript.

  • Are there any examples of radiotherapy at recurrence? Also, if you have researched radiotherapy, you may want to include it in the discussion,

This is an important comment as it points out potential further management in patients with recurrent Rathke’s cleft cysts. We did not experience this in a patient of the presented series. Nevertheless, we discussed this aspect in the revised version of the manuscript.

Round 2

Reviewer 3 Report

The authors have extensively debated all the topics mentioned in the previous revision and fully explained. 

Reviewer 4 Report

The authors have adequately responded to the reviewers' requests.

I believe that the content is worthy of acceptance.